# Gene-specific correlation of RNA and protein levels in human cells and tissues

Fredrik Edfors[1], Frida Danielsson[1], Björn M Hallström[1], Lukas Käll[1], Emma Lundberg[1], Fredrik Pontén[2], Björn Forsström[1] & Mathias Uhlén[1,3,*]

## Abstract

**An important issue for molecular biology is to establish whether transcript levels of a given gene can be used as proxies for the corresponding protein levels. Here, we have developed a targeted proteomics approach for a set of human non-secreted proteins based on parallel reaction monitoring to measure, at steady-state conditions, absolute protein copy numbers across human tissues and cell lines and compared these levels with the corresponding mRNA levels using transcriptomics. The study shows that the transcript and protein levels do not correlate well unless a gene-specific RNA-to-protein (RTP) conversion factor independent of the tissue type is introduced, thus significantly enhancing the predictability of protein copy numbers from RNA levels. The results show that the RTP ratio varies significantly with a few hundred copies per mRNA molecule for some genes to several hundred thousands of protein copies per mRNA molecule for others. In conclusion, our data suggest that transcriptome analysis can be used as a tool to predict the protein copy numbers per cell, thus forming an attractive link between the field of genomics and proteomics.**

**Keywords** gene expression; protein quantification; targeted proteomics; transcriptomics

**Subject Categories** Genome-Scale & Integrative Biology; Post-translational Modifications, Proteolysis & Proteomics; Transcription

**Mol Syst Biol. (2016) 12: 883**

See also: **GM Silva and C Vogel** (October 2016)

## Introduction

Fundamental biological processes govern the flow of information from genome to gene product to cellular phenotype (Payne, 2015). The correlation between mRNA levels and the corresponding protein levels is in this context an important issue, and the presence or absence of such correlation on an individual gene/protein level has been debated in literature for many years (Anderson & Seilhamer, 1997; Gry *et al*, 2009; Maier *et al*, 2009, 2011; Lundberg & Uhlén, 2010; Schwanhäusser *et al*, 2011; Lawless *et al*, 2016). Resolving these conflicting reports is of fundamental interest for both genome and proteome research, since massive efforts to characterize the steady-state transcriptome in various human cells and tissues are ongoing, including the HPA (Uhlén *et al*, 2015), GTEx consortium (Melé *et al*, 2015), and ENCODE (ENCODE Project Consortium *et al*, 2012) efforts. If RNA levels could be used to predict protein levels, the value of these extensive expression resources would substantially increase, thereby allowing protein level prediction studies based on genomewide transcriptomics data tremendously benefit systems biology efforts of human biology and disease. However, numerous reports have concluded (Nagaraj *et al*, 2011; Vogel & Marcotte, 2012; Payne, 2015) that proteome and transcriptome abundances are not sufficiently correlated to act as proxies for each other. In contrast, several recent reports based on genome-scale data have suggested a correlation between the steady-state levels of mRNA indicating a constant protein–mRNA ratio in human cell lines (Lundberg *et al*, 2010) and tissues (Wilhelm *et al*, 2014). This led to the hypothesis that protein abundance in any given tissue might be predicted from mRNA abundance (Wilhelm *et al*, 2014). These conflicting results thus call for more in-depth studies to clarify this issue.

Here, we decided to investigate the correlation using a targeted proteomics approach with internal standards to allow the determination of the absolute copy number of molecules across human cell lines and tissues, in contrast to previous studies based on label-free absolute quantification of proteins that have been shown to underestimate proteins over large dynamic ranges (Ahrné *et al*, 2013). The targeted proteomics method rely on spike-in of known amounts of stable isotope-labeled protein fragments (Zeiler *et al*, 2012) followed by trypsin digestion and parallel reaction monitoring (PRM) analysis (Gallien *et al*, 2012) to determine relative amounts of peptides from sample and internal standard, thereby creating precise anchoring points for all quantitative measurements between all replicates and thus minimizing technical artifacts. Absolute protein copy numbers in the sample can subsequently be calculated from the ratio measured between sample and standard peptides. In contrast to

1 Science for Life Laboratory, KTH – Royal Institute of Technology, Stockholm, Sweden
2 Department of Immunology, Genetics and Pathology, Rudbeck Laboratory, Uppsala University, Uppsala, Sweden
3 Novo Nordisk Foundation Center for Biosustainability, Technical University of Denmark, Hørsholm, Denmark
  *Corresponding author. Tel: +46 70 5132101; E-mail: mathias.uhlen@scilifelab.se

similar methods using labeled peptides as standards, such as AQUA peptides (Gerber *et al*, 2003), the protein fragments are digested simultaneously together with the target protein, which minimizes errors arising during sample preparation, such as the effect of incomplete trypsin digestion or sample loss prior addition of standard.

The protein copy numbers of selected genes were determined across tissues and cell lines representing cells of different origin, and the transcript levels corresponding to the protein-coding genes were established by genomewide transcriptome analysis. This allowed us, for the first time, to compare absolute protein copy numbers per cell with transcript levels measured as TPM (transcripts per million) (Bray *et al*, 2016). An important part of the study was to develop a precise cell count method based on a histone-based normalization procedure to allow the absolute number of cells be established also for complex tissue samples containing mixtures of cell types. Based on this normalization and the precise determination of protein copy numbers, we demonstrate that the predictability of the protein copy numbers from RNA levels can be significantly enhanced if a gene-specific, cell independent RNA-to-protein (RTP) conversion factor is introduced.

## Results

### Selection of genes and development of PRM assays

The RNA and protein levels were studied in samples from nine human cell lines (Table EV1) and 11 human tissues representing diverse functional units, such as liver, lung, kidney, and tonsil

(Table EV2). The transcriptome of these samples was determined using digital counting of the transcript using RNA-Seq (Mortazavi *et al*, 2008). The number of transcripts per gene was determined as transcript per million (TPM), thus calculating the number of estimated mRNA molecules for a given gene per million of total mRNA molecules in the cell, allowing for a straightforward comparison of transcription levels between samples of different sequencing depths and cell counts. Based on transcript analysis, genes for targeted proteomics analysis were selected based according to the following criteria: (i) intracellular or membrane bound protein product (i.e., non-secreted), (ii) present across most of the analyzed tissues and cells, and (iii) having a relatively high degree of variability in the analyzed tissues and cells. This resulted in 55 genes suitable for PRM analysis with available protein standards. Transcriptomics data across the cell lines and tissues for these genes are shown in Table EV3.

To allow for a precise determination of copy number of the corresponding proteins, PRM assays were developed (Table EV4) representing each of the 55 genes with stable isotope-labeled recombinant protein fragments (QPrESTs) produced in a bacterial host and quantified as described before (Zeiler *et al*, 2012). PRM assays, in most cases based on at least two independent peptides, were developed (Table EV5), and the sample-specific concentration of isotope-labeled standard to be spiked-in to reach approximately one-to-one ratio between standard and endogenous target protein were determined using lysates from a selection of cell lines (U2OS and HEK293). This allowed us to assemble a multiplex mixture of 69 isotope-labeled QPrEST standards, some genes covered by multiple standards, with the concentration of each standard

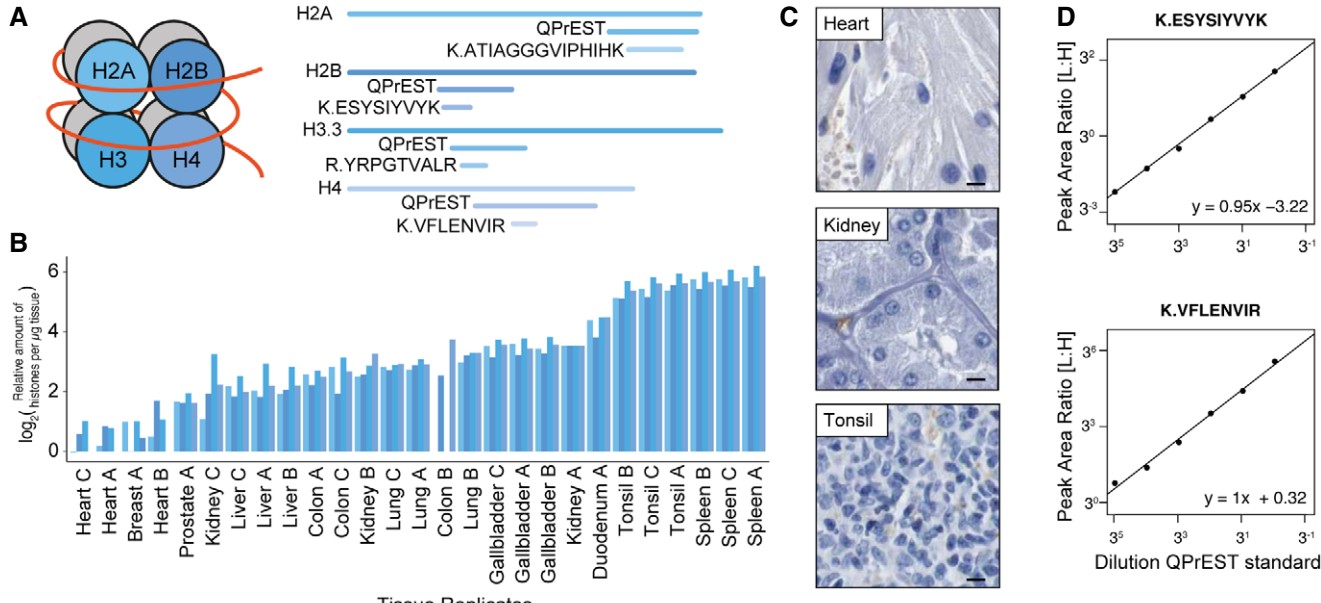

**Figure 1.  Determination of cell counts using the histone abundance for normalization.**

A   The core histones and overview of the corresponding QPrEST and peptide standards mapped out on the protein sequence.
B   Relative quantification of all four histone proteins in each tissue replicate (order of appearance per replicate: H2A, H2B, H3.3, and H4).
C   Immunohistochemistry images from the Human Protein Atlas (http://www.proteinatlas.org) for protein ANXA1 with nuclear staining (blue) for three selected tissues (scale bars = 100 μm).
D   Calibration curves for two of the four histone peptides, with decreasing amount of QPrEST standard spiked into a U2OS cell lysate.

reflecting the abundance of the corresponding protein targets in the cell lines. The assembly of this QPrEST mixture allowed us to perform multiplex analysis of all the 55 protein targets using targeted mass spectrometry.

### Normalization of tissue samples using PRM-based histone quantification

To analyze the number of cells in the tissue samples, we took advantage of the QPrEST approach to develop a quantitative assay based on the four core histone subunits (H2A, H2B, H3, and H4) (Fig 1A). Histones have previously been shown to give a good estimate of DNA content in various samples using label-free approaches (Wiśniewski *et al*, 2014), and here, we introduce isotope-labeled recombinant QPrEST standards in all our assays representing the four major histones. An analysis of cell numbers present in the different tissue samples (Fig 1B) showed that there are many more cells per mg tissue from spleen and tonsil as

compared to heart. This observation is supported by immunohistochemistry (Fig 1C) showing many more cells with nuclear staining in tonsil as compared to heart muscle. The number of proteins quantified in each tissue sample was therefore normalized based on the histones and subsequently used to calculate cell counts for each tissue in this study, as shown in Table EV6. Dilution series of these standards demonstrated a good linearity (Fig 1D) based on an assay using the heavy standard spiked into a serial dilution of a U2OS cell lysate.

### The protein copy number of the target genes in tissues and cell lines

Using the multiplex QPrEST mixture, the protein copy number for the 55 target proteins was determined in all the samples. The results for all cell lines and tissues are summarized in Table EV8, and examples of the results are summarized in Fig 2A. The protein levels for the various target proteins ranged from thousands to

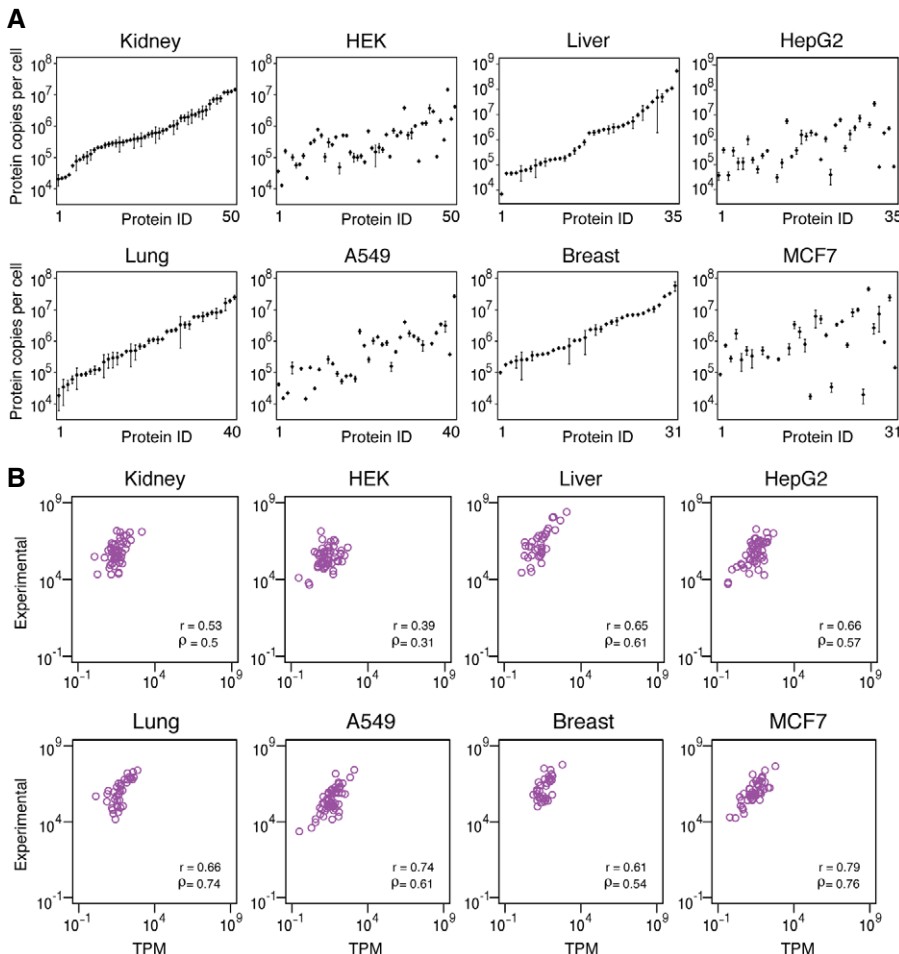

**Figure 2. Absolute copy number of proteins in tissues and corresponding cell lines.**

A   Absolute copy number of protein in kidney tissue and human embryonic kidney cells (HEK293), liver tissue and liver cancer cell line (HepG2), lung tissue and lung cancer cell line (A549), and breast tissue and breast cancer cell line (MCF7). The order of proteins is the same in the tissue and corresponding cell line, and the proteins have been ordered according to the abundance in the respective tissue.

B   The direct correlation between RNA (TPM) and protein abundances (copy number) for all quantified genes in the same tissues and cell lines. Spearman's (ρ) and Pearson's (*r*) correlation between the two values across the quantified genes are shown. The other seven tissues and five cell lines are shown in Fig EV4.

hundreds of millions of copies per cell. As an example, the absolute copy number per average cell in the kidney ranged from 20,500 protein molecules for a nucleotidase (CANT1) to 15 million for a leukocyte elastase inhibitor (SERPINB1). Interestingly, the absolute copy numbers per cell of many of the target proteins are significantly different in the kidney-derived cell line HEK293 demonstrating, as noted earlier (Uhlén *et al*, 2015), that caution should be taken to use cell lines as models for normal tissue. This observation is supported also when comparing the absolute copy number of proteins in liver and the liver-derived cell line HepG2, the lung and the lung-derived cell line A549, and the breast and the breast-derived cell line MCF7.

### The direct correlation between RNA and protein levels in tissues and cell lines

We then decided to compare directly the RNA and protein levels of the target genes in the different tissues and cell lines. In Fig 2B, the RNA levels and protein copy number for the analyzed genes are plotted for some of the cell lines and tissues. A moderate correlation can be observed, and this is reflected in calculation of the Pearson's correlations across the genes. The Pearson's correlation range from 0.39 in the kidney-derived HEK293 to 0.79 in the breast-derived cell line MCF7 with a correlation around 0.6 for all the tissues. These results are in line with earlier results (Schwanhäusser *et al*, 2013) showing a moderate correlation when RNA and protein levels are compared directly without taking gene-specific differences into account.

### The gene-specific correlation of RNA and protein levels for selected genes

Next, the gene-specific RNA-to-protein correlation was investigated for each gene separately. In Fig 3, some examples of the protein copy number and the RNA levels (TPM) are shown across the nine cell lines and the 11 tissues. For each gene, the correlation between RNA and protein levels across the cells and tissues was calculated as Spearman's (rho) or Pearson's ($r$ or $R^2$) correlations. The similarity of the ratio between RNA and protein levels across the cells of different origin allowed us also to calculate an average RNA-to-protein (RTP) ratio independent of cellular origin. As an example, selenium binding protein 1 (SELENBP1) shows a similar pattern of expression between RNA and protein levels across the samples and this is confirmed by a high Pearson's correlation ($r = 0.90$, log-log) resulting in an average RTP ratio of 220,000. Similarly, stomatin (STOM) shows a high correlation ($r = 0.89$), but with a much lower average RTP ratio of 26,000. The third example, argininosuccinate synthetase 1 (ASS1), also shows a high correlation ($r = 0.89$) with a slightly higher RTP ratio (32,500). In Fig EV1, the RNA and protein levels for all the 55 genes are shown and the Pearson's and Spearman's correlations with the average RTP ratios are summarized in Table EV7. The gene-specific RNA-to-protein conversion factor is shown for all genes and samples, and in Fig 4A, the RTP ratio across the nine cell lines and eleven tissues are summarized as box-plots to visualized the variation of RTP values between the samples, but also between different genes. The analysis suggests that the RTP ratios are relatively constant for an individual gene independent of origin of cell and tissue, although the ratio differs significantly between the genes with the RTP ratios varying from 200 for a transcription factor (MYBL2) to 220,000 for SERPINB1, most likely

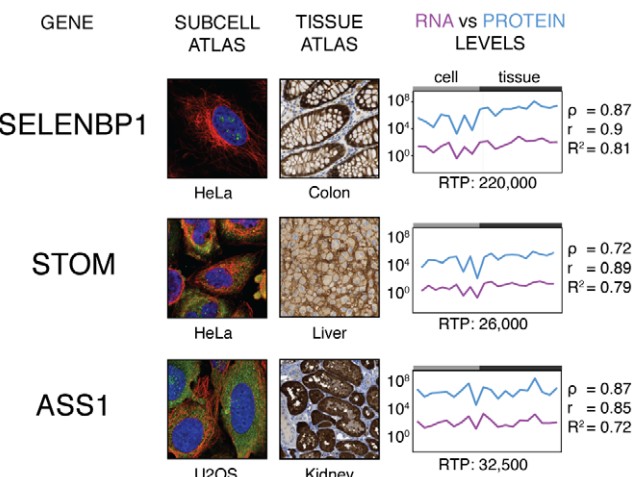

**Figure 3.** The protein and RNA levels for three genes.
Subcellular localization by immunofluorescence staining and immunohistochemistry staining in tissue sections by three different antibodies (SELENBP1, HPA011731; STOM, HPA010961; ASS1, HPA020896). Microtubule and nuclear probes are visualized in red and blue, respectively. Antibody staining is shown in green. RNA-to-protein ratio across nine cell lines and 11 tissues with Spearman's ρ, Pearson's *r* and $R^2$ for each gene. All other genes can be found in Fig EV1.

reflecting differences in translation rate and/or protein degradation for individual proteins. In Fig EV2A, the coefficient of variation of the RTP ratios is plotted versus protein length showing a tendency for higher variation for longer proteins across the analyzed samples, although general statements must be verified with analysis of more genes in the future.

In Fig EV2B, the RTP ratios are plotted versus protein length showing a tendency for higher RTP ratios for smaller sized proteins, although the generality of this must be further investigated by including more genes in the analysis. Interestingly, an analysis of the RTP ratios for proteins in different cellular compartments (Fig EV3) suggests that there are subcellular effects. As an example, higher RTP ratios are in general observed for proteins in the extracellular space. Again, this tendency must be further investigated with more genes before general statements can be made.

### Prediction of protein copy number based on RTP ratios

The results above suggest that protein copy number can be roughly predicted from the corresponding RNA levels using a gene-specific RTP ratio independent of cellular origin. Thus, the mean RNA values in each tissue and cell were multiplied with the gene-specific RNA-to-protein conversion factor and the protein copy numbers predicted from the RNA values were plotted against the experimentally determined protein copy number for all the genes for some of the tissues and cell lines (Fig 4B). Note that in each case, the gene-specific RNA-to-protein conversion factor used for prediction of protein copy number was calculated from the other nineteen cells and tissues, excluding the plotted tissue in order to avoid overfitting. As shown, a good correlation can be observed across all the genes in each of the tissues and cells suggesting that the RNA levels can be used to predict the corresponding protein copy number per cell using the gene-specific RTP ratio (Figs EV4 and EV5).

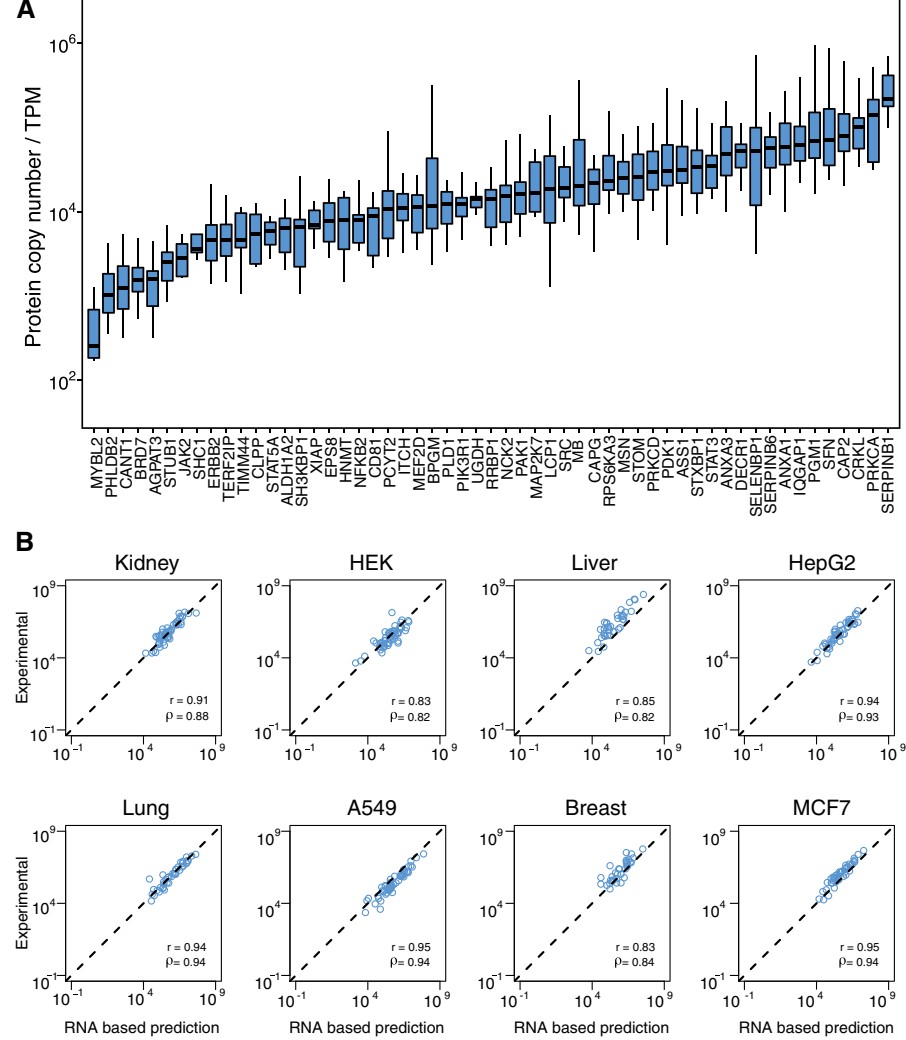

**Figure 4. The correlation between the absolute copy number of proteins and the corresponding RNA levels (measured as TPM) in nine cell lines and 11 tissues.**

A   The gene-specific RNA-to-protein correlation factors are shown for all the 55 genes with a box-plot showing the average correlation factor for each gene and the variation observed in the nine cell lines and 11 tissues. All the values for each of the cell lines and tissues are found in Table EV7. Horizontal lines = median. The lower and upper "hinges" correspond to the first and third quartiles (the 25th and 75th percentiles). Length of the whiskers as multiple of IQR = 1.5.

B   The gene-specific correlation between protein copy number (*x*-axis) and predicted protein copy number based on the RNA levels (RNA-based prediction) is shown for four tissues and four cell lines. The other seven tissues and five cell lines are also shown in Fig EV5 and predicted copy numbers can be found in Table EV9.

The robustness of the prediction was assessed by varying the number of samples used for prediction of the RNA-to-protein conversion factor (Fig 5A). The results show that the conversion factor calculated from four or more random samples (training set), predicting all other samples (test set), yields a median Pearson's correlation higher than 0.9. These results suggest that it is enough to determine the RTP ratio in a few cell lines or tissues and then used the mean to determine a "universal" gene-specific RTP ratio to predict protein copy number across other cells and tissues.

The correlation between RNA and protein levels for all the analyzed genes in the various cell lines and tissues was plotted based on Pearson's correlation to allow a summary comparison before and after the use of the gene-specific RTP correlation factor (Fig 5B). The Pearson's correlations vary significantly across the cell lines and tissues when a direct comparison is carried out with a medium correlation of 0.67. This correlation is significantly enhanced when the gene-specific RTP ratio is applied for each protein to yield a median Pearson's correlation of 0.93. An overview of these results is shown in Fig 5C, in which the obtained Pearson's correlations over the 55 genes in the nine cell lines and eleven tissues are plotted with and without using the gene-specific RTP-conversion factor. A clear improvement of predictability is obtained by introducing the gene-specific RTP ratio.

## Discussion

The evidence that genomewide transcriptomics data can be used as proxies for the corresponding steady-state protein copy numbers in cells and tissues has far-reaching consequences and thus

      

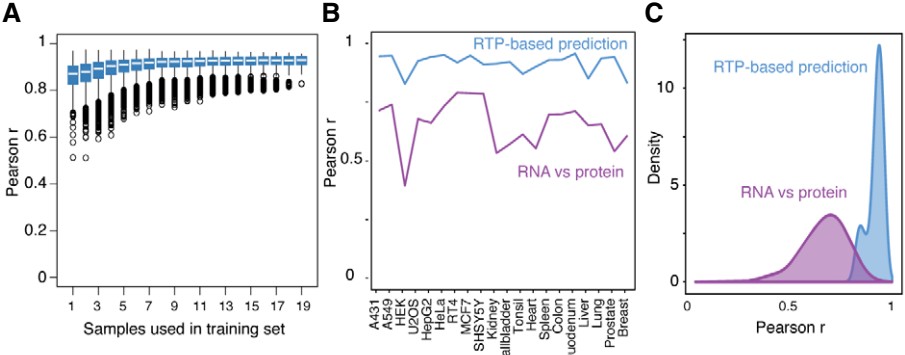

**Figure 5. The gene-specific correlation between RNA and protein levels.**

A   The gene-specific correlation between protein copy number (*x*-axis) and predicted protein copy number based on the RNA levels (RNA-based prediction) is shown for all the 55 genes and all 20 cell lines and tissues. Horizontal lines = median. The lower and upper "hinges" correspond to the first and third quartiles (the 25th and 75th percentiles). Length of the whiskers as multiple of IQR. Defaults to 1.5. Circles indicate outliers.

B   The Pearson's correlation between RNA and protein levels for the 55 genes in the nine cell lines and 11 tissues is shown as a direct comparison of RNA and protein levels (purple, RNA versus protein) and after introducing the gene-specific correlation factor (blue, RNA-based prediction versus protein).

C   Density plot for the direct comparison between RNA and protein levels before and after introducing the RTP-conversion factor. The Pearson's correlation using the RTP-conversion factor is improved substantially for all cell lines and tissues with a median Pearson's correlation of 0.93.

justifies the use of genomewide transcriptomics data for molecular studies involving the analysis of protein levels, including metabolic modeling, systems biology, and biomarker discovery efforts. The results reinforce the importance of open source transcriptomics data resources, such as GTEx (Melé *et al*, 2015), HPA (Uhlén *et al*, 2015), FANTOM (FANTOM Consortium and the RIKEN PMI and CLST (DGT), 2014), TCGA (Cancer Genome Atlas Research Network *et al*, 2013), and Expression Atlas (Petryszak *et al*, 2016), and the data confirm that these resources are valuable also for researchers interested in analyzing protein abundances. The results support a strategy for genomewide expression studies with an initial analysis of transcript levels using transcriptomics followed by a more in-depth analysis of the relevant protein products using direct protein analysis, such as antibody-based methods or mass spectrometry.

In order to determine the steady-state number of protein molecules per cell, it is important to be able to establish the exact number of cells in a sample, which is relatively straightforward by cell counting of *in vitro* cultivated cells, but more challenging for tissues lysates as the heterogeneity of cells present varies across different tissues. Here, we took advantage of the QPrEST approach to develop a quantitative assay based on the four core histone subunits (H2A, H2B, H3, and H4) known to be distributed approximately equally along the chromosomes (van Holde, 1989; Thomas, 1999). In this way, it was possible to calculate the number of cells in the various samples and to normalize each tissue with regard to the presence of number of cells per mg of sample. An analysis of the tissue samples showed that there are many more cells per mg tissue from spleen and tonsil than compared to the heart, with 30 times more histone protein per weight tissue. Noteworthy, the determined ratio between individual histone genes is conserved across tissue types, suggesting that the level of modification in the quantified region is relatively conserved (the regions were originally selected to show few possible modification sites as reported by Uniprot), which strengthens the quantification method as each histone can be used as control for the others. Each tissue sample was thus

normalized to allow the number of cells to be approximately calculated from each tissue sample, thereby eliminating artifacts that arise from interference by proteins from the extracellular matrix or by differences in cell size.

The data presented here demonstrate that the predictability of protein copy numbers from RNA levels can be significantly enhanced whether a gene-specific, cell and tissue independent RNA-to-protein (RTP) conversion factor is introduced and the results from normalization of the tissues are taken into account. The results show that the RTP ratio varies hugely between different genes suggesting that one mRNA molecule in some cases can generate close to a million protein copies at steady state, while mRNA from other genes generate in average less than thousand proteins under the same conditions. This is not surprising, since it known that protein half-lives can vary many orders of magnitude and that proteins also have different translational rates (Schwanhäusser *et al*, 2011; Vogel & Marcotte, 2012). However, our data imply that these gene-specific differences in RNA-to-protein ratio are independent of cell or tissue, and thus, a "universal" RTP ratio can be determined that can be used across cells of different origin and stage for a given gene product. Although a relatively limited amount of genes have been investigated here, the results support the view that the translational rates and protein half-lives are roughly the same for a given protein across various cells and tissues. The analysis here was designed to cover intracellular proteins present in a majority of the selected cell and tissue samples, thereby excluding proteins potentially expressed at very low levels in only a fraction of the investigated tissues.

Here, we have analyzed the RTP ratio in cell lines and tissues representing some of the major organs and tissues in the human body, but obviously more work is needed to extend this analysis also to other tissues and to include more genes into the analysis to possibly establish the generality of this observation and to identify exceptions in which RNA and protein levels do not correlate. Due to the inherent issues in the use of RNA as proxies for protein levels, it is important to follow up the RNA-based analysis with protein profiling to conform the results from the transcript analysis. In this context,

more in-depth analysis of factors that might give miss-leading ratios are encouraged, such as the presence of protein modifications on the target protein that will affect the protein copy number determinations and the presence of non-coding RNAs that might affect the transcriptomics analysis. It is also important to point out that the tissues analyzed here consist of mixtures of cell types of different origin and thus only yields the average mRNA and protein levels at steady-state conditions across all the different cell types in the tissue samples.

In summary, our results suggest that the predictability of protein copy numbers from RNA levels can be significantly enhanced if a gene-specific, cell independent RNA-to-protein (RTP) conversion factor is used. Thus, transcriptome analysis can be used as a powerful tool to predict the corresponding protein copy numbers, forming an attractive link between the field of genomics and proteomics. We suggest that the gene-specific RNA-to-protein protein conversion factor should be determined across all protein-coding genes to provide a basic resource for the medical and life science community.

# Materials and Methods

## Ethical statement

Human tissue samples used for protein and mRNA expression analyses were collected and handled in accordance with Swedish laws and regulation and obtained from the Department of Pathology, Uppsala University Hospital, Uppsala, Sweden, as part of the sample collection governed by the Uppsala Biobank (http://www.uppsala biobank.uu.se/en/). All human tissue samples used in the present study were anonymized in accordance with approval and advisory report from the Uppsala Ethical Review Board [Reference # 2002-577, 2005-338 and 2007-159 (protein) and # 2011-473 (RNA)], and consequently, the need for informed consent was waived by the ethics committee.

## Selection of genes

A selection of 60 genes was initially chosen for the study. Firstly, genes coding for predicted secreted proteins were excluded based on a majority decision-based method for secreted proteins (MDSEC) used for protein classification within the Protein Atlas (http://www.proteinatlas.org/humanproteome/secretome#prediction). Secondly, the gene had to be differentially expressed on transcript level across nine cell lines (A431, HepG2, A549, HeLa, HEK293, A549, RT4, MCF7, and SH-SY5Y) subjected for the study. Finally, QPrEST standard had to be available, yielding at least one proteotypic peptide as the protein was degraded into peptides by trypsin.

## Production and quantification of protein standards for absolute quantification

An *Escherichia coli* strain auxotrophic for the amino acids arginine and lysine (Matic *et al*, 2011) was used for recombinant production of heavy isotope-labeled QPrEST standards. DNA fragments were initially cloned into the expression pAff8c (Larsson, 2000) and were thereafter transformed into an *E. coli* strain for recombinant protein production. Cells containing expression vectors were cultivated in 10 ml minimal media using 100-ml shake flasks as previously described (Studier, 2005; Tegel *et al*, 2009). Heavy isotope-labeled ($^{13}$C and $^{15}$N) versions of lysine and arginine (Cambridge Isotope Laboratories, Tewksbury, MA, USA) were provided to the cells at 200 μg/ml to generate fully incorporated heavy protein standards. Cell cultures were harvested, and the QPrESTs were purified using the N-terminal quantification tag (QTag) that included a hexahistidine tag used for immobilize metal ion affinity chromatography (IMAC). After purification, all isotopic QPrEST fragments were absolutely quantified by mass spectrometry against a non-labeled ultra purified QTag-standard, which previously had been quantified by amino acid analysis. The QTag-standard, also including a C-terminal a OneStrep tag, was purified using IMAC chromatography, and the IMAC elution buffer was exchanged for 1× PBS (10 mM NaP, 150 mM NaCl, pH 7.3) using a PD-10 desalting column (GE Healthcare, Uppsala, Sweden). The sample was purified on a StrepTrap HP column (GE Healthcare) on an Äkta explorer system (GE Healthcare) according to the manufactures protocol. All QPrESTs were quantified by mixing 1:1 with QTag-standard and thereafter digested using an in-solution trypsin digestion protocol. Proteins were first reduced with 10 mM dithiothreitol (DTT) for 30 min at 56°C and thereafter followed by addition of 50 mM iodoacetamide (IAA) and incubated in dark for 20 min. Proteomics grade porcine trypsin (Sigma) was added in a 1:50 enzyme to substrate (E:S) ratio and incubated in a thermomixer at 37°C. After 16 h, the reaction was quenched by addition of FA and the sample was desalted using in-house prepared StageTips packed with Empore C18 Bonded Silica matrix (3M, Saint Paul, MN) (Rappsilber *et al*, 2007). Briefly, three layers of octadecyl membrane were placed in 200-μl pipette tips. The membrane was activated by addition of 100% ACN, followed by centrifugation for 1 min at 840 *g*. The membrane was equilibrated by addition of 0.1% FA, MQ followed by centrifugation for 1 min at 840 *g*. The sample was acidified prior addition onto the membrane, followed by centrifugation for 1 min at 840 *g*. The membrane was washed twice with 0.1% FA, MQ, and the peptides were eluted in two steps using 60% ACN, MQ. Desalted peptides were vacuum-dried before subjected for LC-MS analysis.

## Preparation of cell pellets

Nine different cell lines (A431, HepG2, A549, HeLa, HEK 293, U2OS, RT4, MCF7, and SH-SY5Y) were cultivated at 37°C in a humidified atmosphere containing 5% $CO_2$. HEK-293, MCF7, HeLa, and HepG2 were cultivated in Minimum Essential Medium Eagle (Sigma-Aldrich, St Louis, MO, USA). A549 and SH-SY5Y were cultivated in Dulbecco's modified Eagle's medium (Sigma-Aldrich). U2OS and RT4 were cultivated in McCoy's medium (Sigma-Aldrich), and A431 was cultivated in RPMI-1640 (Sigma-Aldrich). All media were supplemented with 10% fetal bovine serum (Sigma-Aldrich). Media for HEK 293, MCF7, HeLa, and HepG2 were supplemented with 1% MEM non-essential amino acid solution (Sigma-Aldrich), and media for MCF7 and HepG2 were also supplemented with 1% L-glutamine (Sigma-Aldrich). The cells were cultivated up to 80% confluence and counted with a Scepter 2.0 Cell Counter (Merck Millipore, Billerica, MA, USA) before pellets were collected and stored at −80°C.

## RNAseq analysis

For the cell lines, RNA was extracted from the cells using the RNEasy® kit (Qiagen), generating high-quality total RNA (i.e.,

RIN > 8) that was used as input material for library construction with Illumina TruSeq Stranded mRNA reagents. The samples were sequenced on the Illumina HiSeq2500 platform to a depth of ~20 million reads. Raw sequences were mapped to the human reference genome GrCh38 and further quantified using the Kallisto software (Bray *et al*, 2016). TPM values for genes were generated by summing up TPM values for the corresponding transcripts generated by Kallisto. All cell line data are available at http://www.ncbi.nlm.nih.gov/bioproject/PRJNA183192.

Procedures for extraction of RNA from tissues, library preparation, and sequencing have been described elsewhere (Uhlén *et al*, 2015). Briefly, reads were mapped to the human reference genome assembly GRCh38 and quantified using Kallisto version 0.42.4. Normalized expression levels (TPM values) on gene level were obtained by summing the estimated values from the constituent transcripts of each gene, respectively. All tissue data are available at http://www.ebi.ac.uk/arrayexpress/experiments/E-MTAB-1733/.

### Cell lysis

Cells were dissolved in lysis buffer (100 mM Tris–HCl, 4% SDS, 10 mM DTT, pH 7.6) and incubated at 95°C in a thermomixer for 5 min at 30 *g* and thereafter sonicated at 50% amp (1 s pulse, 1 s hold) for 1 min.

### Tissue lysis

Twenty consecutive sections from 11 different fresh-frozen human tissues (Table EV2) were subjected for analysis. Tissue sections were disrupted directly from their frozen state by 3-mm tungsten carbide beads using a Tissue Lyser LT (Qiagen, Hilden, Germany) set to maximum speed for 2 min. After complete tissue disruption, 250 µl lysis buffer (100 mM Tris–HCl, 4% SDS, 10 mM DTT, pH 7.6) was added and samples were immediately incubated in a thermomixer for 5 min at 95°C and mixed at 30 *g*. All samples were sonicated for 1 min at 50% amplitude (1 s pulse + 1 s hold) and all clarified by centrifugation at 13,570 *g* for 10 min.

### Filter-aided Sample preparation

One QPrEST mastermix was prepared to represent a 1:1 (L:H) peptide ratio to the endogenous levels in U2OS and HEK293, and the same amount of the mastermix was spiked-in also to all other samples, either to 1 million cells or 600 µg of clarified tissue lysate. The lysate was diluted with denaturing buffer (8 M urea, 100 mM Tris–HCl pH 8.5) and centrifuged through a 0.22-µm spin filter (Corning, Corning, NY, USA). Trypsin digestion was performed using the previously described filter-aided sample preparation (FASP) method (Wiśniewski *et al*, 2009) After overnight digestion using porcine trypsin (Sigma) in a 1:50 E:S ratio, all peptides were extracted from cell line digests and desalted using the same in-house prepared C18 StageTip protocol as described above. Peptides from tissue digests (excluding kidney) were all extracted using strong cation exchange material due to polymers present in the cryopreservative surrounding the fresh-frozen tissue. Briefly, three layers of strong cation matrix (3M, Saint Paul, MN) were placed in 200-µl pipette tips. The membrane was activated by addition of 100%

MeOH, followed by centrifugation for 1 min at 840 *g*. The membrane was equilibrated by addition of wash buffer (30% MeOH, 0.1% FA, MQ) followed by centrifugation for 1 min at 840 *g*. The sample was acidified prior being added onto the membrane, followed by centrifugation for 1 min at 840 *g*. The membranes were washed twice with wash buffer, and peptides were then eluted in two steps using elution buffer (33% NH$_4$OH, 30% MeOH, MQ). Desalted peptides were vacuum-dried before LC-MS analysis.

### Liquid chromatography

Liquid chromatography was performed using an UltiMate 3000 binary RS nano system (Thermo Scientific) with an EASY-Spray ion source. All samples were stored in their lyophilized state and resuspended by the autosampler prior injection as 1 µg sample material was loaded onto a Acclaim PepMap 100 trap column (75 µm × 2 cm, C18, 3 µm, 100 Å), washed 5 min at 0.250 µl/min with solvent A (95% H$_2$O, 5% DMSO, 0.1% FA), and thereafter separated using a PepMap 800 C18 column (15 cm × 75 µm, 3 µm). The gradient went from solvent A to solvent B (90% ACN, 5% H$_2$O, 5% DMSO, 0.1% FA) at a constant flow of 0.250 µl/min, up to 43% solvent B in 40 min, followed by an increase up to 55% in 10 min and thereafter a steep increase to 100% B in 2 min. Online LC–MS was performed using a Q-Exactive HF mass spectrometer (Thermo Scientific).

### Spectral library generation

A pool of 71 QPrESTs representing 60 genes were pooled in equimolar amounts and digested by trypsin according to the in-solution protocol described above. QPrEST peptides were resuspended in 3% ACN, 0.1% FA, MQ prior LC-MS analysis, and 50 fmol per QPrEST-ID was injected onto column. A Top5 MS-method with master scans performed at 60,000 resolution (mass range 300–1,600 *m/z*, AGC 3e6) was followed by five consecutive MS2 at 30,000 resolution (AGC 1e5, underfill ratio 0.1%) with normalized collision energy set to 25. Raw files were searched using MaxQuant (Cox & Mann, 2008), using the search engine Andromeda against QPrEST sequences (Table EV4) with an *E. coli* (BL21 Uniprot-ID: #UP000002032) background, which was used for recombinant protein production in order to limit false-positive hits against QPrEST peptides. Identified peptides were further processed by only allowing proteotypic peptides mapping to one single human gene (defined by SwissProt), also excluding peptides with miscleavages and peptides including methionines.

### Data-independent MS acquisition

Full MS master scans at 60,000 resolution (mass range 300–1,600 *m/z*, AGC 1e6) were followed by 20 data-independent acquisitions MS/MS at 60,000 resolution (AGC 1e6) defined by a scheduled parallel reaction monitoring (PRM) method (Table EV5). Precursors were isolated with a 1.2 *m/z* isolation window, and maximum injection time was set to 105 ms for both MS1 and MS2, which resulted in a duty cycle of 2.7 s. The isolation list was split into two consecutive LC runs, targeting 120 paired light and heavy peptides per injection.

## MS-data evaluation and protein quantification

Raw MS-files (available at: http://www.proteinatlas.org/download/prm_cells_tissues.zip) from the data-independent method were processed using Skyline Proteomics Environment (MacLean *et al*, 2010). The ratio between endogenous and heavy peptide standard was calculated from the summed area intensity over the retention time for each peptide fragment separately. Here, five genes were excluded from the analysis as endogenous peptides could not be successfully quantified. All peptide ratios, in each replicate separately, were normalized against the amount of histones quantified in the replicate (Table EV6; Fig EV6) in order account for quantification errors that arise from differences in number of cells subjected for analysis, extracellular matrix in tissue samples, and pipetting errors when spiking in standards. Median peptide ratios between replicates were used to calculate the absolute amount of peptide concentration after normalizing for the absolute and known amount of protein standard that was spiked to each sample. If more than one peptide assay per protein was available (13 genes with two peptides, 11 genes with three peptides, five genes with four peptides, two genes with five peptides), the median peptide value was used for calculation of protein concentration for each replicate.

## RNA-to-protein conversion factor

Protein values were used to calculate a gene-specific RNA-to-protein conversion factor by dividing the amount of protein in each sample by the TPM value for that gene in the corresponding sample. The gene-specific median of all ratios was used to predict theoretical protein levels from the RNA level (TPM), that is, RNA-based prediction, excluding the sample being predicted when calculating the RNA-to-protein conversion factor from all other 19 samples. In order to assess the predictive power of the conserved RNA-to-protein conversion factor across all investigated sample types, different sizes of test and validation sets were used in a k-fold cross-validation as 5,000 protein predictions were made for each test set, ranging from 1 to 19 randomly assigned sample combinations in the training set, predicting all other samples in the test set.

## Equations

Protein amount is dependent on cell size. This calls for a method that controls the number of cells present in an analyzed sample, especially when tissue samples as cell counter cannot be used:

$$\text{Cellular protein number} = \text{Total protein concentration} \times \text{cell volume} \tag{1}$$

Protein copies per cells is given by:

$$\text{Protein number per cell} = \frac{\text{Total protein number}}{\text{Number of cells}} \tag{2}$$

DNA amount is a proxy for number of cells (Milo, 2013) as the amount of DNA per 2N human cell equals approximately 3.6 pg:

$$\text{Number of cells} \approx \frac{\text{Total DNA mass}}{3.59 \text{ pg DNA per cell}} \tag{3}$$

Also, DNA and histones are good proxies for each other as the number of histones per cell is proportional toward amount of DNA per cell, that is, same for all 2N human cells (Wiśniewski *et al*, 2014) (note: not applicable for cell lines with different karyotypes):

$$\text{Number of histones} \propto \text{DNA length} \tag{4}$$

$$\text{Number of cells} = \frac{\text{Number of histones}}{\text{Number of histones per cell}} \tag{5}$$

This is applicable for all 2N human cells and also intra-cell lines with the same karyotype (i.e., technical replicates):

$$\text{Protein number per cell} \propto \frac{\text{Total protein amount}}{\text{Number of histones}} \tag{6}$$

**Expanded View** for this article is available online.

## Acknowledgements

We acknowledge the entire staff of the Human Protein Atlas program and the Science for Life Laboratory for valuable contributions. We thank Jens Nielsen, Per-Åke Nygren, and Adil Mardinoglu for valuable comments and advice. We thank the Department of Pathology at the Uppsala Akademiska hospital, Uppsala, Sweden, and Uppsala Biobank for providing tissue specimens used in this study. Funding was provided by the Knut and Alice Wallenberg Foundation and Erling Persson Foundation to MU Correspondence and requests for materials should be addressed to MU.

## Author contributions

MU designed the study. FE and BF performed the laboratory work. FE, FD MU, LK, and BMH did the bioinformatics and statistical analysis. MU, FE, and BF wrote the manuscript with contributions from EL and FP.

## Conflict of interest

The authors declare that they have no conflict of interest.

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
