## [Review Process File · Molecular Systems Biology]

Gene-specific correlation of RNA and protein levels in human cells and tissues

Fredrik Edfors, Frida Danielsson, Björn M. Hallström, Lukas Käll, Emma Lundberg, Fredrik Pontén, Björn Forsström and Mathias Uhlén

Corresponding author: Mathias Uhlén, KTH - Royal Institute of Technology

Review timeline:

Submission date:	05 July 2016
Editorial Decision:	01 August 2016
Revision received:	08 August 2016
Editorial Decision:	26 August 2016
Revision received:	05 September 2016
Accepted:	15 September 2016

Editor: Maria Polychronidou

Transaction Report:

1st Editorial Decision

01 August 2016

Thank you again for submitting your work to Molecular Systems Biology. We have now heard back from two of the three referees who agreed to evaluate your study. Unfortunately, after a couple of reminders we have not yet received a report from reviewer #3. Since the recommendations of the other two referees are similar, I prefer to make a decision now rather than further delaying the process. As you will see below, the reviewers acknowledge that you address an important topic. However, they raise a number of concerns, which should be carefully addressed in a revision of the manuscript.

The referees' recommendations are quite clear, so there is no need to repeat all the points listed below. One of the more fundamental issues refers to the need to include further analyses on the protein-RNA relationships for different genes. Reviewer #1 (point 1) provides constructive suggestions and as s/he points out, such analyses would significantly enhance the impact of the study.

REFeree COMMENTS

Reviewer #1:

The authors present a new view on the long-standing discussion on the relationship between protein and mRNA concentrations: the variation of the relationship for individual genes across mammalian

tissues. To do so, precise concentration measurements are taken for 55 genes across tissues and cell lines. The manuscript is well-written and the idea is solid. It is of broad interest and therefore suitable for a journal like Molecular Systems Biology.

However, before supporting publication of the work, I would like to see several criticisms addressed.

MAJOR

1. The novel aspect of the work is the assessment of the protein-to-RNA relationship for a given gene ACROSS tissues. The finding is very interesting, but is, in my view, still underrepresented in the current version of the m/s. Much of the figures/results/discussion is about correlation of protein and mRNA within tissues/cell lines - that has been looked at before and many times. The novelty of this work lies in the plots in Figure 3 and should, in my opinion, be much more analyzed. For example: which genes have an extremely constant protein-to-RNA relationship, which vary? What would the functions be of these genes, are there any general conclusions regarding to which genes vary and which don't? Is it perhaps correlated with their overall abundance? I.e. highly abundant genes vary less (since it's harder)? Is there perhaps a certain tissue/cell type in which the correlation is consistently off? In Figure 3 for the three genes, the 3rd tissue from the right seems to be consistently different. What is the reason for this? Technical? Or perhaps cells in this tissue are arrested in a specific cell cycle stage and therefore the histone normalization is thrown off? Or, at a per gene basis - is there evidence for additional post-transcriptional regulation in a specific tissue for genes where the protein-to-mRNA relationship deviates from the average in one case? What could be the biology behind this? I would urge the authors to go deeper this route of analysis.

2. Relatedly - I would like to have the presentation/discussion MUCH more turned around the biology behind this. I don't think protein-per-RNA ratios and a prediction factor are that interesting, what is much more biologically relevant is the fact that the gross translation/protein degradation rate appears to be set at a per-gene basis (perhaps due to sequence, length, etc properties of the gene) and does not vary across tissues. That is, the order of magnitude in translation/protein degradation of a gene is constant. However, smaller changes (two-fold etc) still exist across tissues, confirming hypotheses drawn from many other studies that suggest that post-transcriptional regulation FINE-TUNES gene expression levels (see recent reviews, e.g. in Cell by Aebersold or earlier in Nature Rev Genetics by Marcotte). The work presented here is consistent with this and adds another dimension.

3. Relatedly - it might be nice to cite Uri Alon et al.'s work on Fold-Change-Detection in bacterial chemotaxis. It seems related to this whole discussion and a nice new twist.

4. The dynamic range of concentrations of RNA or protein cover 3-4 or 5 orders of magnitude. That is ok, but still only a small part of what is seen. In particular low abundance proteins seem to be missing in the analysis, and it needs to be discussed what is expected for them. Perhaps this relationship (see above) does not hold true for them, also since it is easier for the cell to change the concentration of low-abundance proteins. Limitations of the findings need to be discussed. - On a related note, it needs to be discussed to what extent the currently selected proteins are representative in their expression nature.

5. In Table S8, the PTR numbers for A549, HepG2, HeLa, MCF7, SHSY5Y for LCP1 gene are 56, 158, 138, 430, 245. These are 5 out of 20 tissue/cell line in total with the values less than $5 \cdot 10^2$. However, in the Figure 4A shows a minimum larger than 10^4 . The authors need to explain and justify why/how these lower values have been left out and that this is not cherry picking.

6. The normalization based on histones is one way to normalize for abundance, but it can have biases: some tissues might have cells arrested in a specific cell cycle stage etc etc. At least for some extreme cases (see other suggestions), I would strongly suggest to validate with alternative assessments. The DNA content can be measured per ug tissue, the total protein and RNA content too. The number of cells per ug or ul can also be estimated, at least for some tissues. Since much of the conclusions rely on this normalization, it needs to be rock-solid.

MINOR

1. There is no Figure 2A in the document.
2. The axes labels of Figure 2 are cut. The axis ranges should also be adjusted to not show so much empty space.
3. In Figure 3, what the different colors are representing in the immunofluorescence staining? The immunohistochemistry staining needs to be explained in more detail.
4. Figure 5 is at low resolution and hardly legible.
5. In Table S8, second column, the authors mention "Order in Fig. 1B" - what does that mean?

Reviewer #2:

This manuscript describes the use of PRM data to test how well transcript levels correlate with protein abundance across tissues.

1. Differing conclusions. Conclusion described in Abstract is decidedly unexciting and certainly not novel but this really under-sells the real conclusion from the data - that transcript and protein levels do not correlate very well at all unless one has a gene-specific correction factor.
2. Absolute copy number per cell - the authors could do more to explain why knowing this is important, apart from just having more knowledge. They make a big deal out of it but it seems like a sidebar to their main purpose of correlating RNA and protein levels.

Articles (of the grammatical variety) and prepositions missing in several instances, particularly preceding numbers (e.g., in Abstract it should be "to close to A million copies", in Results it should be "almost hundredS OF millions of copies")

1st Revision - authors' response

08 August 2016

We are happy that the reviewers found our manuscript is well-written and of broad interest. We find the issues and comments raised by reviewers relevant and we have prepared a revised manuscript taken these suggestions into account.

In the following are point-to-point comments regarding the issues brought up by the reviewers:

Reviewer #1 comments:

Reviewer: 1. The novel aspect of the work is the assessment of the protein-to-RNA relationship for a given gene ACROSS tissues. The finding is very interesting, but is, in my view, still underrepresented in the current version of the m/s. Much of the figures/results/discussion is about correlation of protein and mRNA within tissues/cell lines - that has been looked at before and many times. The novelty of this work lies in the plots in Figure 3 and should, in my opinion, be much more analyzed. For example: which genes have an extremely constant protein-to-RNA relationship, which vary? What would the functions be of these genes, are there any general conclusions regarding to which genes vary and which don't? Is it perhaps correlated with their overall abundance? I.e. highly abundant genes vary less (since it's harder)? Is there perhaps a certain tissue/cell type in which the correlation is consistently off? In Figure 3 for the three genes, the 3rd tissue from the right seems to be consistently different. What is the reason for this? Technical? Or perhaps cells in this tissue are arrested in a specific cell cycle stage and therefore the histone normalization is thrown off? Or, at a per gene basis - is there evidence for additional post-transcriptional regulation in a specific tissue for genes where the protein-to-mRNA relationship deviates from the average in one case? What could be the biology behind this? I would urge the

authors to go deeper this route of analysis.

Comment: This is a relevant comment and we have extended the analysis regarding Figure 3 considerably. It is important to note that we do not want to make too much generalized statements based on the relative small number of genes analyzed, but we have extended the analysis of the determined RTP-ratios both in terms of protein length and subcellular localization (Including two new Supplementary Figures).

Reviewer: 2. Relatedly - I would like to have the presentation/discussion MUCH more turned around the biology behind this. I don't think protein-per-RNA ratios and a prediction factor are that interesting, what is much more biologically relevant is the fact that the gross translation/protein degradation rate appears to be set at a per-gene basis (perhaps due to sequence, length, etc properties of the gene) and does not vary across tissues. That is, the order of magnitude in translation/protein degradation of a gene is constant. However, smaller changes (two-fold etc) still exist across tissues, confirming hypotheses drawn from many other studies that suggest that post-transcriptional regulation FINE-TUNES gene expression levels (see recent reviews, e.g. in Cell by Aebersold or earlier in Nature Rev Genetics by Marcotte). The work presented here is consistent with this and adds another dimension.

Comment: Again, a relevant comment and we have extended the discussion to include more discussions on subcellular localization. For example, proteins localized to the extracellular space and centrosome have higher RTP-ratios. In contrast, proteins annotated and associated to the nucleolus have lower RTP ratios.

Reviewer: 3. Relatedly - it might be nice to cite Uri Alon et al.'s work on Fold-Change-Detection in bacterial chemotaxis. It seems related to this whole discussion and a nice new twist.

Comment: The paper by Alon et al regarding bacterial chemotaxis is indeed very interesting, but the topic quite unrelated to the present work. We agree that a discussion on the mechanism of bacterial chemotaxis might give an extended view, but it is hard to give this observation justice without a lengthy discussion and we do not want to reach too far away from the scope of our work.

Reviewer: 4. The dynamic range of concentrations of RNA or protein cover 3-4 or 5 orders of magnitude. That is ok, but still only a small part of what is seen. In particular low abundance proteins seem to be missing in the analysis, and it needs to be discussed what is expected for them. Perhaps this relationship (see above) does not hold true for them, also since it is easier for the cell to change the concentration of low-abundance proteins. Limitations of the findings need to be discussed. - On a related note, it needs to be discussed to what extent the currently selected proteins are representative in their expression nature.

Comment: This is an interesting point. We have added a few sentences regarding this to the Discussion.

Reviewer: 5. In Table S8, the PTR numbers for A549, HepG2, HeLa, MCF7, SHSY5Y for LCP1 gene are 56, 158, 138, 430, 245. These are 5 out of 20 tissue/cell line in total with the values less than $5 \cdot 10^2$ 500. However, in the Figure 4A shows a minimum larger than 10^4 . The authors need to explain and justify why/how these lower values have been left out and that this is not cherry picking.

Comment: This has been corrected.

Reviewer: 6. The normalization based on histones is one way to normalize for abundance, but it can have biases: some tissues might have cells arrested in a specific cell cycle stage etc etc. At least for some extreme cases (see other suggestions), I would strongly suggest to validate with alternative assessments. The DNA content can be measured per μg tissue, the total protein and RNA content too. The number of cells per μg or μl can also be estimated, at least for some tissues. Since much of the conclusions rely on this normalization, it needs to be rock-solid.

Comment: We agree, but it is not easy to do experimentally due to the uncertainty to related DNA or RNA amounts to the samples size analyzed on the mass spectrometry instrument. We believe that the use of internal standards for absolute quantification of histones further improves the previous work on histone normalization performed by Wisniewski et al.

MINOR

Reviewer: 1. There is no Figure 2A in the document.

Comment: The wrong Figure 2 was submitted. Figure 2A is now included in the revised manuscript.

Reviewer: 2. The axes labels of Figure 2 are cut. The axis ranges should also be adjusted to not show so much empty space.

Comment: See previous comment.

Reviewer: 3. In Figure 3, what the different colors are representing in the immunofluorescence staining? The immunohistochemistry staining needs to be explained in more detail.

Comment: More explanation has been included in the figure legend.

Reviewer: 4. Figure 5 is at low resolution and hardly legible.

Comment: A new Figure 5 has been submitted with high resolution images.

Reviewer: 5. In Table S8, second column, the authors meantion "Order in Fig. 1B" - what does that mean?

Comment: The explanation has been clarified.

Reviewer #2 comments:

Reviewer: 1. Differing conclusions. Conclusion described in Abstract is decidedly unexciting and certainly not novel but this really under-sells the real conclusion from the data - that transcript and protein levels do not correlate very well at all unless one has a gene-specific correction factor.

Comment: The abstract has been revised.

Reviewer: 2. Absolute copy number per cell - the authors could do more to explain why knowing this is important, apart from just having more knowledge. They make a big deal out of it but it seems like a sidebar to their main purpose of correlating RNA and protein levels.

Comment: The Discussion regarding this has been extended.

Reviewer: 3. Articles (of the grammatical variety) and prepositions missing in several instances, particularly preceding numbers (e.g., in Abstract it should be "to close to A million copies", in Results it should be "almost hundredS OF millions of copies")

Comments: these two typographical errors have been corrected.

2nd Editorial Decision

26 August 2016

Thank you for submitting your revised manuscript to Molecular Systems Biology. We have now heard back from reviewer #1 who was asked to evaluate the study. As you will see below, s/he thinks that the study has been improved. However, s/he lists some remaining concerns, which we

would ask you to address in a minor revision.

While we think that inclusion of qPCR data as an independent validation of the RNA-seq data is not mandatory, we would not object to the inclusion of such data i.e. in case they are already available. In line with the comments of reviewer #1, we would ask you to include a more detailed explanation of the histone normalization approach and to extend the description/discussion on the findings related to the relationship between the conversion factor and protein function, length etc. Also, we agree with reviewer #1 that it should be explicitly mentioned in the abstract that the gene-specific conversion factor is independent of the tissue-type.

 REFEREE COMMENTS

Reviewer #1:

Edfors et al. present a revised manuscript in which they addressed many of the reviewers' comments. The text is clearer to read and has a consistent story. However: while I do like the scope of the work and I consider the main findings important for the field and a broader readership, but I disagree with the authors on the emphasis and presentation of the results.

Because of this discrepancy, I list reasons below for and against publication of the work in its current form:

PROS (no particular order)

1. Protein concentration measurements are of highest-quality. While many studies have examined the protein-vs-mRNA question, having such a good dataset can finally exclude some technical biases.
2. The 55 proteins/mRNAs span a wide range of concentrations, and have been selected based on their variation in mRNA expression across tissues. Both very nice and interesting findings.
3. The number of tissues examined is larger than in other studies.
4. The normalization (using histones) is as thorough and high-quality as it can be.
5. The finding that the protein-per-mRNA ratio is set by gene TYPE rather than by TISSUE is relevant for the field, as it means that, when measuring mRNA and estimating protein concentrations from that, it is sufficient to know an approximate conversion factor, as this study shows that the conversion factor is relatively constant across tissues.

CONS (no particular order)

1. The RNA seq data does not have orthogonal validation (e.g. qPCR).
2. The histone normalization is not compared to alternative approaches. Given that in principle, the authors do the same thing as the Nature paper 2014 (ref 16), they *really* need to explain and demonstrate why their current dataset is of much higher quality.
3. Presentation/analysis still has flaws:
 1. Typos and redundancy in use of terms/words.
 2. The 55 genes were selected based on them being intracellular, but the enriched category in the RTP analysis is EXTRA-cellular proteins. Explain/discuss?
 4. Insufficient analysis of the results (see above). I do not think it is enough to report a conversion factor - its interpretation with respect to its variability across gene functions, tissues, gene length, protein abundance is part of the discussion. I had several suggestions in my first review, and the authors examined function (discussed briefly, but not interpreted) and length (only in Supplement). I am not so happy with how the authors followed-through with it. The new version of the manuscript has some good interpretation of the result, but I think it's still hidden. E.g p. 9 "However, our data implies that these gene-specific differences in RNA to protein ratio are independent of cell or tissue and thus a "universal" RTP-ratio can be determined that can be used across cells of different origin and stage for a given gene product. "
 5. Along the lines of comment 4, Abstract and Introduction still focus on solving the debate on protein-vs-RNA correlation, and I think the paper doesn't really answer that question. The current version of the paper basically says "protein and mRNA correlate better across genes if a gene-

specific conversion factor is applied". But so what? I find it much more interesting that this gene-specific conversion factor is somewhat constant across tissues and therefore gene TYPE is the major determinant for protein-vs-mRNA ratio rather than tissue. Minor note: With that result in mind, an analysis of a biased set of 55 genes is absolutely fine (and the authors don't have to apologize for using few genes).

6. Minor: apart from spiking in the peptides before tryptic digest, what is 'novel' or 'new' about the PRM method? What *is* truly new is that it has been used for many genes and many tissues in a consistent study.

2nd Revision - authors' response

05 September 2016

We are pleased with the comments and suggestions both by the reviewer and the editor. We find the suggestions relevant and we have prepared a revised manuscript taken these suggestions into account. During uploading of all files, we re-examined all calculations and found a small error in the use of one of the software algorithms used in the Skyline package calculating the protein standards. This only affects the quantification of our standards and do not affect the cross-tissue and cell line RTP-analysis, but this affects the absolute numbers reported in the manuscript. Therefore, we have now updated the tables and figures accordingly.

Reviewer #1 comments:

1. The RNA seq data does not have orthogonal validation (e.g. qPCR).

Several studies have shown a correlation between qPCR and RNA-Seq and this was not the scope of this investigation. Examples of such publications are: Su, Łabaj, Li et al. (2014) A comprehensive assessment of RNA-seq accuracy, reproducibility and information content by the sequencing Quality control consortium, *Nature Biotechnology* and Nagalakshmi et al, (2008) The Transcriptional Landscape of the Yeast Genome Defined by RNA Sequencing, *Science*).

2. The histone normalization is not compared to alternative approaches. Given that in principle, the authors do the same thing as the Nature paper 2014 (ref 16), they *really* need to explain and demonstrate why their current dataset is of much higher quality.

This has been addressed in the introduction as we included a relevant reference to Ahrné et al (2003).

3. Presentation/analysis still has flaws:

1. Typos and redundancy in use of terms/words.
2. The 55 genes were selected based on them being intracellular, but the enriched category in the RTP analysis is EXTRA-cellular proteins. Explain/discuss?

Extra-cellular is membrane-bound in this sense, and we have clarified this in the revised manuscript by changing the wording to "non-secreted" instead of intracellular.

4. Insufficient analysis of the results (see above). I do not think it is enough to report a conversion factor - its interpretation with respect to its variability across gene functions, tissues, gene length, protein abundance is part of the discussion. I had several suggestions in my first review, and the authors examined function (discussed briefly, but not interpreted) and length (only in Supplement). I am not so happy with how the authors followed-through with it. The new version of the manuscript has some good interpretation of the result, but I think it's still hidden. E.g p. 9 "However, our data implies that these gene-specific differences in RNA to protein ratio are independent of cell or tissue and thus a "universal" RTP-ratio can be determined that can be used across cells of different origin and stage for a given gene product. "

We are hesitant to include a more in-depth discussion on protein function since the number of genes in each category is rather limited.

5. Along the lines of comment 4, Abstract and Introduction still focus on solving the debate on protein-vs-RNA correlation, and I think the paper doesn't really answer that question. The current version of the paper basically says "protein and mRNA correlate better across genes if a gene-specific conversion factor is applied". But so what? I find it much more interesting that this gene-specific conversion factor is somewhat constant across tissues and therefore gene TYPE is the major determinant for protein-vs-mRNA ratio rather than tissue. Minor note: With that result in mind, an analysis of a biased set of 55 genes is absolutely fine (and the authors don't have to apologize for using few genes).

We agree.

6. Minor: apart from spiking in the peptides before tryptic digest, what is 'novel' or 'new' about the PRM method? What **is** truly new is that it has been used for many genes and many tissues in a consistent study.

This has been updated in the manuscript. More emphasis on the novelty of this approach that is comparing genes across several cell lines and tissues, quantified by spiked in internal standards.

With these changes in the revised manuscript, I hope you find the manuscript suitable for publication in Molecular Systems Biology.

MOLECULAR SYSTEMS BIOLOGY

Corresponding Author Name: Mathias Uhlén

Manuscript Number: MSB-16-7144RR